# K-Medoids for K-Means Seeding

**James Newling**
Idiap Research Institue and
École polytechnique fédérale de Lausanne
`james.newling@idiap.ch`

**François Fleuret**
Idiap Research Institue and
École polytechnique fédérale de Lausanne
`francois.fleuret@idiap.ch`

## Abstract

We show experimentally that the algorithm `clarans` of Ng and Han (1994) finds better $K$-medoids solutions than the Voronoi iteration algorithm of Hastie et al. (2001). This finding, along with the similarity between the Voronoi iteration algorithm and Lloyd's $K$-means algorithm, motivates us to use `clarans` as a $K$-means initializer. We show that `clarans` outperforms other algorithms on 23/23 datasets with a mean decrease over `k-means-++` (Arthur and Vassilvitskii, 2007) of $30\%$ for initialization mean squared error (MSE) and $3\%$ for final MSE. We introduce algorithmic improvements to `clarans` which improve its complexity and runtime, making it a viable initialization scheme for large datasets.

## 1   Introduction

### 1.1   $K$-means and $K$-medoids

The $K$-means problem is to find a partitioning of points, so as to minimize the sum of the squares of the distances from points to their assigned partition's mean. In general this problem is NP-hard, and in practice approximation algorithms are used. The most popular of these is Lloyd's algorithm, henceforth `lloyd`, which alternates between freezing centers and assignments, while updating the other. Specifically, in the *assignment* step, for each point the nearest (frozen) center is determined. Then during the *update* step, each center is set to the mean of points assigned to it. `lloyd` has applications in data compression, data classification, density estimation and many other areas, and was recognised in Wu et al. (2008) as one of the top-10 algorithms in data mining.

The closely related $K$-medoids problem differs in that the center of a cluster is its medoid, not its mean, where the medoid is the cluster member which minimizes the sum of *dissimilarities* between itself and other cluster members. In this paper, as our application is $K$-means initialization, we focus on the case where dissimilarity is squared distance, although $K$-medoids generalizes to non-metric spaces and arbitrary dissimilarity measures, as discussed in §SM-A.

By modifying the update step in `lloyd` to compute medoids instead of means, a viable $K$-medoids algorithm is obtained. This algorithm has been proposed at least twice (Hastie et al., 2001; Park and Jun, 2009) and is often referred to as the Voronoi iteration algorithm. We refer to it as `medlloyd`.

Another $K$-medoids algorithm is `clarans` of Ng and Han (1994, 2002), for which there is no direct $K$-means equivalent. It works by randomly proposing swaps between medoids and non-medoids, accepting only those which decrease MSE. We will discuss how `clarans` works, what advantages it has over `medlloyd`, and our motivation for using it for $K$-means initialization in §2 and §SM-A.

### 1.2   $K$-means initialization

`lloyd` is a *local* algorithm, in that far removed centers and points do not directly influence each other. This property contributes to `lloyd`'s tendency to terminate in poor minima if not well initial-

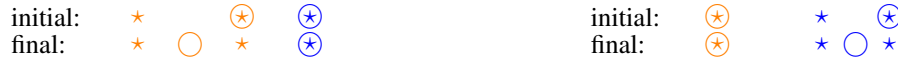

Figure 1: $N = 3$ points, to be partitioned into $K = 2$ clusters with `lloyd`, with two possible initializations (top) and their solutions (bottom). Colors denote clusters, stars denote samples, rings denote means. Initialization with `clarans` enables jumping between the initializations on the left and right, ensuring that when `lloyd` eventually runs it avoids the local minimum on the left.

ized. Good initialization is key to guaranteeing that the refinement performed by `lloyd` is done in the vicinity of a good solution, an example showing this is given in Figure 1.

In the comparative study of $K$-means initialization methods of Celebi et al. (2013), 8 schemes are tested across a wide range of datasets. Comparison is done in terms of speed (time to run initialization+`lloyd`) and energy (final MSE). They find that 3/8 schemes should be avoided, due to poor performance. One of these schemes is uniform initialization, henceforth `uni`, where $K$ samples are randomly selected to initialize centers. Of the remaining 5/8 schemes, there is no clear best, with results varying across datasets, but the authors suggest that the algorithm of Bradley and Fayyad (1998), henceforth `bf`, is a good choice.

The `bf` scheme of Bradley and Fayyad (1998) works as follows. Samples are separated into $J$ $(= 10)$ partitions. `lloyd` with `uni` initialization is performed on each of the partitions, providing $J$ centroid sets of size $K$. A superset of $JK$ elements is created by concatenating the $J$ center sets. `lloyd` is then run $J$ times on the superset, initialized at each run with a distinct center set. The center set which obtains the lowest MSE on the superset is taken as the final initializer for the final run of `lloyd` on all $N$ samples.

Probably the most widely implemented initialization scheme other than `uni` is `k-means++` (Arthur and Vassilvitskii, 2007), henceforth `km++`. Its popularity stems from its simplicity, low computational complexity, theoretical guarantees, and strong experimental support. The algorithm works by sequentially selecting $K$ seeding samples. At each iteration, a sample is selected with probability proportional to the square of its distance to the nearest previously selected sample.

The work of Bachem et al. (2016) focused on developing sampling schemes to accelerate `km++`, while maintaining its theoretical guarantees. Their algorithm `afk-mc`[2] results in as good initializations as `km++`, while using only a small fraction of the $KN$ distance calculations required by `km++`. This reduction is important for massive datasets.

In none of the 4 schemes discussed is a center ever replaced once selected. Such refinement is only performed during the running of `lloyd`. In this paper we show that performing refinement during initialization with `clarans`, before the final `lloyd` refinement, significantly lowers $K$-means MSEs.

## 1.3 Our contribution and paper summary

We compare the $K$-medoids algorithms `clarans` and `medlloyd`, finding that `clarans` finds better local minima, in §3 and §SM-A. We offer an explanation for this, which motivates the use of `clarans` for initializing `lloyd` (Figure 2). We discuss the complexity of `clarans`, and briefly show how it can be optimised in §4, with a full presentation of acceleration techniques in §SM-D.

Most significantly, we compare `clarans` with methods `uni`, `bf`, `km++` and `afk-mc`[2] for $K$-means initialization, and show that it provides significant reductions in initialization and final MSEs in §5. We thus provide a conceptually simple initialization scheme which is demonstrably better than `km++`, which has been the de facto initialization method for one decade now.

Our source code at `https://github.com/idiap/zentas` is available under an open source license. It consists of a C++ library with Python interface, with several examples for diverse data types (sequence data, sparse and dense vectors), metrics (Levenshtein, $l_1$, etc.) and potentials (quadratic as in $K$-means, logarithmic, etc.).

## 1.4 Other Related Works

Alternatives to `lloyd` have been considered which resemble the swapping approach of `clarans`. One is by Hartigan (1975), where points are randomly selected and reassigned. Telgarsky and

Vattani (2010) show how this heuristic can result in better clustering when there are few points per cluster.

The work most similar to `clarans` in the $K$-means setting is that of Kanungo et al. (2002), where it is indirectly shown that `clarans` finds a solution within a factor 25 of the optimal $K$-medoids clustering. The local search approximation algorithm they propose is a hybrid of `clarans` and `lloyd`, alternating between the two, with sampling from a kd-tree during the `clarans`-like step. Their source code includes an implementation of an algorithm they call 'Swap', which is exactly the `clarans` algorithm of Ng and Han (1994).

## 2 Two $K$-medoids algorithms

Like `km++` and `afk-mc`$^2$, $K$-medoids generalizes beyond the standard $K$-means setting of Euclidean metric with quadratic potential, but we consider only the standard setting in the main body of this paper, referring the reader to SM-A for a more general presentation. In Algorithm 1, `medlloyd` is presented. It is essentially `lloyd` with the update step modified for $K$-medoids.

| **Algorithm 1** two-step iterative `medlloyd` algorithm (in vector space with quadratic potential). | **Algorithm 2** swap-based `clarans` algorithm (in a vector space and with quadratic potential). |
|---|---|
| 1: Initialize center indices $c(k)$, as distinct elements of $\{1, \ldots, N\}$, where index $k \in \{1, \ldots, K\}$.<br>2: **do**<br>3:     **for** $i = 1 : N$ **do**<br>4:         $a(i) \leftarrow \underset{k \in \{1, \ldots, K\}}{\arg\min} \|x(i) - x(c(k))\|^2$<br>5:     **end for**<br>6:     **for** $k = 1 : K$ **do**<br>7:         $c(k) \leftarrow$<br>8:         $\underset{i:a(i)=k}{\arg\min} \sum_{i':a(i')=k} \|x(i) - x(i')\|^2$<br>9:     **end for**<br>10: **while** $c(k)$ changed for at least one $k$ | 1: $n_r \leftarrow 0$<br>2: Initialize center indices $\mathcal{C} \subset \{1, \ldots, N\}$<br>3: $\psi_- \leftarrow \sum_{i=1}^{N} \min_{i' \in \mathcal{C}} \|x(i) - x(i')\|^2$<br>4: **while** $n_r \leq N_r$ **do**<br>5:     sample $i_- \in \mathcal{C}$ and $i_+ \in \{1, \ldots, N\} \setminus \mathcal{C}$<br>6:     $\psi_+ \leftarrow \sum_{i=1}^{N}$<br>7:         $\min_{i' \in \mathcal{C} \setminus \{i_-\} \cup \{i_+\}} \|x(i) - x(i')\|^2$<br>8:     **if** $\psi_+ < \psi_-$ **then**<br>9:         $\mathcal{C} \leftarrow \mathcal{C} \setminus \{i_-\} \cup \{i_+\}$<br>10:         $n_r \leftarrow 0, \quad \psi_- \leftarrow \psi_+$<br>11:     **else**<br>12:         $n_r \leftarrow n_r + 1$<br>13:     **end if**<br>14: **end while** |

In Algorithm 2, `clarans` is presented. Following a random initialization of the $K$ centers (line 2), it proceeds by repeatedly proposing a random swap (line 5) between a center ($i_-$) and a non-center ($i_+$). If a swap results in a reduction in energy (line 8), it is implemented (line 9). `clarans` terminates when $N_r$ consecutive proposals have been rejected. Alternative stopping criteria could be number of accepted swaps, rate of energy decrease or time. We use $N_r = K^2$ throughout, as this makes proposals between all pairs of clusters probable, assuming balanced cluster sizes.

`clarans` was not the first swap-based $K$-medoids algorithm, being preceded by `pam` and `clara` of Kaufman and Rousseeuw (1990). It can however provide better complexity than other swap-based algorithms if certain optimisations are used, as discussed in §4.

When updating centers in `lloyd` and `medlloyd`, assignments are frozen. In contrast, with swap-based algorithms such as `clarans`, assignments change along with the medoid index being changed ($i_-$ to $i_+$). As a consequence, swap-based algorithms look one step further ahead when computing MSEs, which helps them escape from the minima of `medlloyd`. This is described in Figure 2.

## 3 A Simple Simulation Study for Illustration

We generate simple 2-D data, and compare `medlloyd`, `clarans`, and baseline $K$-means initializers `km++` and `uni`, in terms of MSEs. The data is described in Figure 3, where sample initializations are also presented. Results in Figure 4 show that `clarans` provides significantly lower MSEs than `medlloyd`, an observation which generalizes across data types (sequence, sparse, etc), metrics (Levenshtein, $l_\infty$, etc), and potentials (exponential, logarithmic, etc), as shown in Appendix SM-A.

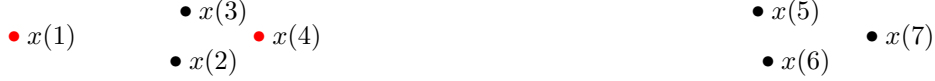

Figure 2: Example with $N = 7$ samples, of which $K = 2$ are medoids. Current medoid indices are 1 and 4. Using medlloyd, this is a local minimum, with final clusters $\{x(1)\}$, and the rest. clarans may consider swap $(i_-, i_+) = (4, 7)$ and so escape to a lower MSE. The key to swap-based algorithms is that cluster assignments are never frozen. Specifically, when considering the swap of $x(4)$ and $x(7)$, clarans assigns $x(2)$, $x(3)$ and $x(4)$ to the cluster of $x(1)$ *before* computing the new MSE.

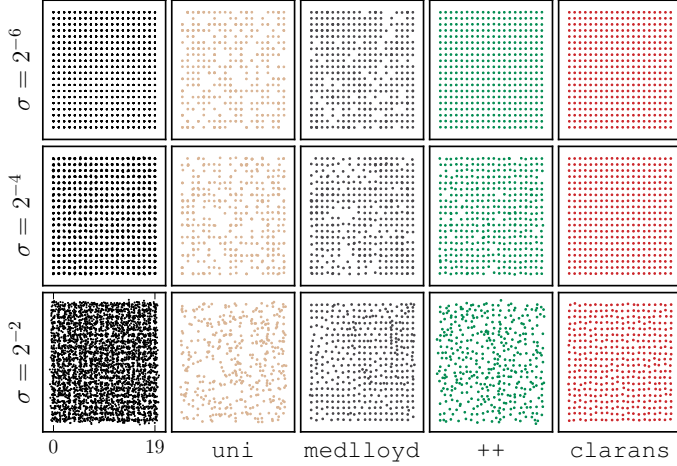

Figure 3: (*Column 1*) Simulated data in $\mathbf{R}^2$. For each cluster center $g \in \{0, \ldots, 19\}^2$, 100 points are drawn from $\mathcal{N}(g, \sigma^2 I)$, illustrated here for $\sigma \in \{2^{-6}, 2^{-4}, 2^{-2}\}$. (*Columns 2,3,4,5*) Sample initializations. We observe 'holes' for methods uni, medlloyd and km++. clarans successfully fills holes by removing distant, under-utilised centers. The spatial correlation of medlloyd's holes are due to its locality of updating.

## 4   Complexity and Accelerations

lloyd requires $KN$ distance calculations to update $K$ centers, assuming no acceleration technique such as that of Elkan (2003) is used. The cost of several iterations of lloyd outweighs initialization with any of uni, km++ and afk-mc$^2$. We ask if the same is true with clarans initialization, and find that the answer depends on how clarans is implemented. clarans as presented in Ng and Han (1994) is $O(N^2)$ in computation and memory, making it unusable for large datasets. To make clarans scalable, we have investigated ways of implementing it in $O(N)$ memory, and devised optimisations which make its complexity equivalent to that of lloyd.

clarans consists of two main steps. The first is swap *evaluation* (line 6) and the second is swap *implementation* (scope of if-statement at line 8). Proposing a good swap becomes less probable as MSE decreases, thus as the number of swap implementations increases the number of consecutive rejected proposals ($n_r$) is likely to grow large, illustrated in Figure 5. This results in a larger fraction of time being spent in the evaluation step.

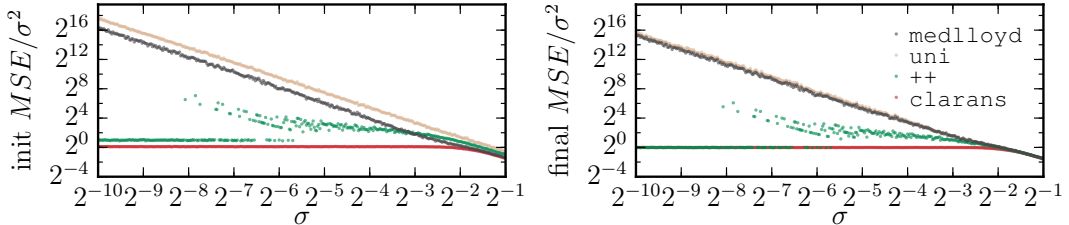

Figure 4: Results on simulated data. For 400 values of $\sigma \in [2^{-10}, 2^{-1}]$, initialization (left) and final (right) MSEs relative to true cluster variances. For $\sigma \in [2^{-5}, 2^{-2}]$ km++ never results in minimal MSE ($MSE/\sigma^2 = 1$), while clarans does for all $\sigma$. Initialization MSE with medlloyd is on average 4 times lower than with uni, but most of this improvement is regained when lloyd is subsequently run (final $MSE/\sigma^2$).

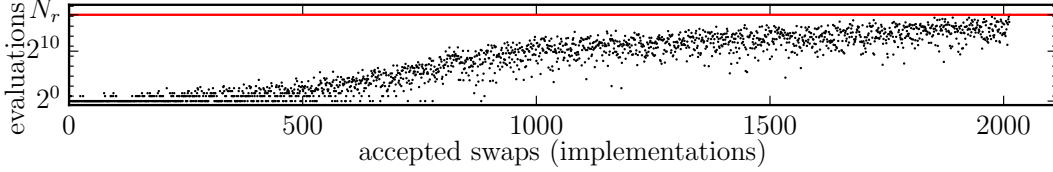

Figure 5: The number of consecutive swap proposal rejections (evaluations) before one is accepted (implementations), for simulated data (§3) with $\sigma = 2^{-4}$.

We will now discuss optimisations in order of increasing algorithmic complexity, presenting their computational complexities in terms of evaluation and implementation steps. The explanations here are high level, with algorithmic details and pseudocode deferred to §SM-D.

**Level -2**   To evaluate swaps (line 6), simply compute all $KN$ distances.

**Level -1**   Keep track of nearest centers. Now to evaluate a swap, samples whose nearest center is $x(i_-)$ need distances to all $K$ samples indexed by $\mathcal{C} \setminus \{i_-\} \cup \{i_+\}$ computed in order to determine the new nearest. Samples whose nearest is not $x(i_-)$ only need the distance to $x(i_+)$ computed to determine their nearest, as either, (1) their nearest is unchanged, or (2) it is $x(i_+)$.

**Level 0**   Also keep track of second nearest centers, as in the implementation of Ng and Han (1994), which recall is $O(N^2)$ in memory and computes all distances upfront. Doing so, nearest centers can be determined for *all* samples by computing distances to $x(i_+)$. If swap $(i_-, i_+)$ is accepted, samples whose new nearest is $x(i_+)$ require $K$ distance calculations to recompute second nearests. Thus from level -1 to 0, computation is transferred from evaluation to implementation, which is good, as implementation is less frequently performed, as illustrated in Figure 5.

**Level 1**   Also keep track, for each cluster center, of the distance to the furthest cluster member as well as the maximum, over all cluster members, of the minimum distance to another center. Using the triangle inequality, one can then frequently eliminate computation for clusters which are unchanged by proposed swaps with just a single center-to-center distance calculation. Note that using the triangle inequality requires that the $K$-medoids dissimilarity is metric based, as is the case in the $K$-means initialization setting.

**Level 2**   Also keep track of center-to-center distances. This allows whole clusters to be tagged as unchanged by a swap, without computing any distances in the evaluation step.

We have also considered optimisations which, unlike levels -2 to 2, do not result in the exact same clustering as `clarans`, but provide additional acceleration. One such optimisation uses random sub-sampling to evaluate proposals, which helps significantly when $N/K$ is large. Another optimisation which is effective during initial rounds is to *not* implement the first MSE reducing swap found, but to rather continue searching for approximately as long as swap implementation takes, thus balancing time between searching (evaluation) and implementing swaps. Details can be found in §SM-D.3.

The computational complexities of these optimisations are in Table 1. Proofs of these complexities rely on there being $O(N/K)$ samples changing their nearest or second nearest center during a swap. In other words, for any two clusters of sizes $n_1$ and $n_2$, we assume $n_1 = \Omega(n_2)$. Using level 2 complexities, we see that if a fraction $p(\mathcal{C})$ of proposals reduce MSE, then the expected complexity is $O(N(1 + 1/(p(\mathcal{C})K)))$. One cannot marginalise $\mathcal{C}$ out of the expectation, as $\mathcal{C}$ may have no MSE reducing swaps, that is $p(\mathcal{C}) = 0$. If $p(\mathcal{C})$ is $O(K)$, we obtain complexity $O(N)$ per swap, which is equivalent to the $O(KN)$ for $K$ center updates of `lloyd`. In Table 2, we consider run times and distance calculation counts on simulated data at the various levels of optimisation.

## 5   Results

We first compare `clarans` with `uni`, `km++`, `afk-mc`[2] and `bf` on the first 23 publicly available datasets in Table 3 (datasets 1-23). As noted in Celebi et al. (2013), it is common practice to run initialization+`lloyd` several time and retain the solution with the lowest MSE. In Bachem et al. (2016) methods are run a fixed number of times, and *mean* MSEs are compared. However, when comparing *minimum* MSEs over several runs, one should take into account that methods vary in their time requirements.

| | -2 | -1 | 0 | 1 | 2 |
|---|---|---|---|---|---|
| 1 evaluation | $NK$ | $N$ | $N$ | $\frac{N}{K} + K$ | $\frac{N}{K}$ |
| 1 implementation | 1 | 1 | $N$ | $N$ | $N$ |
| $K^2$ evaluations, $K$ implementations | $K^3N$ | $K^2N$ | $K^2N$ | $NK + K^3$ | $KN$ |
| memory | $N$ | $N$ | $N$ | $N$ | $N + K^2$ |

Table 1: The complexities at different levels of optimisation of *evaluation* and *implementation*, in terms of required distance calculations, and overall memory. We see at level 2 that to perform $K^2$ evaluations and $K$ implementations is $O(KN)$, equivalent to `lloyd`.

| | -2 | -1 | 0 | 1 | 2 |
|---|---|---|---|---|---|
| $\log_2(\#\,\mathrm{dcs})$ | 44.1 | 36.5 | 35.5 | 29.4 | 26.7 |
| time [s] | - | - | 407 | 19.2 | 15.6 |

Table 2: Total number of distance calculations ($\#\,\mathrm{dcs}$) and time required by `clarans` on simulation data of §3 with $\sigma = 2^{-4}$ at different optimisation levels.

| dataset | # | N | dim | K | TL [s] |
|---|---|---|---|---|---|
| a1 | 1 | 3000 | 2 | 40 | 1.94 |
| a2 | 2 | 5250 | 2 | 70 | 1.37 |
| a3 | 3 | 7500 | 2 | 100 | 1.69 |
| birch1 | 4 | 100000 | 2 | 200 | 21.13 |
| birch2 | 5 | 100000 | 2 | 200 | 15.29 |
| birch3 | 6 | 100000 | 2 | 200 | 16.38 |
| ConfLong | 7 | 164860 | 3 | 22 | 30.74 |
| dim032 | 8 | 1024 | 32 | 32 | 1.13 |
| dim064 | 9 | 1024 | 64 | 32 | 1.19 |
| dim1024 | 10 | 1024 | 1024 | 32 | 7.68 |
| europe | 11 | 169308 | 2 | 1000 | 166.08 |

| dataset | # | N | dim | K | TL [s] |
|---|---|---|---|---|---|
| housec8 | 12 | 34112 | 3 | 400 | 18.71 |
| KDD* | 13 | 145751 | 74 | 200 | 998.83 |
| mnist | 14 | 10000 | 784 | 300 | 233.48 |
| Mopsi | 15 | 13467 | 2 | 100 | 2.14 |
| rna* | 16 | 20000 | 8 | 200 | 6.84 |
| s1 | 17 | 5000 | 2 | 30 | 1.20 |
| s2 | 18 | 5000 | 2 | 30 | 1.50 |
| s3 | 19 | 5000 | 2 | 30 | 1.39 |
| s4 | 20 | 5000 | 2 | 30 | 1.44 |
| song* | 21 | 20000 | 90 | 200 | 71.10 |
| susy* | 22 | 20000 | 18 | 200 | 24.50 |
| yeast | 23 | 1484 | 8 | 40 | 1.23 |

Table 3: The 23 datasets. Column 'TL' is time allocated to run with each initialization scheme, so that no new runs start after TL elapsed seconds. The starred datasets are those used in Bachem et al. (2016), the remainder are available at `https://cs.joensuu.fi/sipu/datasets`.

Rather than run each method a fixed number of times, we therefore run each method as many times as possible in a given time limit, 'TL'. This dataset dependent time limit, given by columns TL in Table 3, is taken as 80× the time of a single run of `km++`+`lloyd`. The numbers of runs completed in time TL by each method are in columns 1-5 of Table 4. Recall that our stopping criterion for `clarans` is $K^2$ consecutively rejected swap proposals. We have also experimented with stopping criterion based on run time and number of swaps implemented, but find that stopping based on number of rejected swaps best guarantees convergence. We use $K^2$ rejections for simplicity, although have found that fewer than $K^2$ are in general needed to obtain minimal MSEs.

We use the fast `lloyd` implementation accompanying Newling and Fleuret (2016) with the 'auto' flag set to select the best exact accelerated algorithm, and run until complete convergence. For initializations, we use our own C++/Cython implementation of level 2 optimised `clarans`, the implementation of `afk-mc`$^2$ of Bachem et al. (2016), and `km++` and `bf` of Newling and Fleuret (2016).

The objective of Bachem et al. (2016) was to prove and experimentally validate that `afk-mc`$^2$ produces initialization MSEs equivalent to those of `km++`, and as such `lloyd` was not run during experiments. We consider both initialization MSE, as in Bachem et al. (2016), and final MSE after `lloyd` has run. The latter is particularly important, as it is the objective we wish to minimize in the $K$-means problem.

In addition to considering initialization and final MSEs, we also distinguish between mean and minimum MSEs. We believe the latter is important as it captures the varying time requirements, and as mentioned it is common to run `lloyd` several times and retain the lowest MSE clustering. In Table 4 we consider two MSEs, namely mean initialization MSE and minimum final MSE.

| | runs completed | | | | | mean initial mse | | | | minimum final mse | | | | |
|---|---|---|---|---|---|---|---|---|---|---|---|---|---|---|
| | km++ | afk mc2 | uni | bf | cla rans | km++ | afk mc2 | uni | cla rans | km++ | afk mc2 | uni | bf | cla rans |
| 1 | 135 | 65 | 138 | 8 | 29 | 1 | 0.97 | 2 | **0.63** | 0.59 | 0.58 | 0.59 | 0.61 | **0.57** |
| 2 | 81 | 24 | 85 | 5 | 7 | 1 | 0.99 | 1.96 | **0.62** | 0.6 | 0.59 | 0.61 | 0.63 | **0.58** |
| 3 | 82 | 21 | 87 | 6 | 4 | 1 | 0.99 | 2.07 | **0.63** | 0.6 | 0.61 | 0.62 | 0.63 | **0.59** |
| 4 | 79 | 27 | 95 | 28 | 5 | 1 | 0.99 | 1.54 | **0.69** | 0.66 | 0.66 | 0.66 | 0.66 | **0.66** |
| 5 | 85 | 22 | 137 | 27 | 6 | 1 | 1 | 3.8 | **0.62** | 0.62 | 0.62 | 0.64 | 0.63 | **0.59** |
| 6 | 68 | 22 | 77 | 23 | 4 | 1 | 0.98 | 2.35 | **0.67** | 0.64 | 0.64 | 0.68 | 0.68 | **0.63** |
| 7 | 84 | 66 | 75 | 38 | 46 | 1 | 1 | 1.17 | **0.73** | 0.64 | 0.64 | 0.64 | **0.64** | 0.64 |
| 8 | 84 | 29 | 88 | 5 | 19 | 1 | 0.98 | 43.1 | **0.65** | 0.65 | 0.65 | 0.66 | 0.66 | **0.63** |
| 9 | 81 | 29 | 90 | 5 | 16 | 1 | 1.01 | $>10^2$ | **0.66** | 0.66 | 0.66 | 0.66 | 0.69 | **0.63** |
| 10 | 144 | 52 | 311 | 24 | 18 | 1 | 0.99 | $>10^2$ | **0.72** | 0.62 | 0.61 | 0.62 | 0.62 | **0.59** |
| 11 | 70 | 25 | 28 | 15 | 4 | 1 | 1 | 20.2 | **0.72** | 0.67 | 0.67 | 2.25 | 2.4 | **0.64** |
| 12 | 80 | 27 | 81 | 21 | 4 | 1 | 0.99 | 2.09 | **0.77** | 0.7 | 0.7 | 0.73 | 0.74 | **0.69** |
| 13 | 102 | 74 | 65 | 56 | 5 | 1 | 1 | 4 | **0.77** | 0.69 | 0.69 | 0.75 | 0.75 | **0.69** |
| 14 | 88 | 43 | 276 | 83 | 4 | 1 | 1 | 1 | **0.87** | 0.6 | 0.6 | 0.6 | 0.61 | **0.6** |
| 15 | 91 | 23 | 52 | 7 | 4 | 1 | 1 | 25 | **0.6** | 0.57 | 0.57 | 3.71 | 3.62 | **0.51** |
| 16 | 107 | 28 | 86 | 28 | 4 | 1 | 0.99 | 24.5 | **0.62** | 0.62 | 0.61 | 2.18 | 2.42 | **0.56** |
| 17 | 84 | 31 | 85 | 5 | 25 | 1 | 1.01 | 2.79 | **0.7** | 0.66 | 0.65 | 0.67 | 0.69 | **0.65** |
| 18 | 100 | 39 | 100 | 7 | 30 | 1 | 0.99 | 2.24 | **0.69** | 0.65 | 0.65 | 0.66 | 0.66 | **0.64** |
| 19 | 88 | 36 | 83 | 6 | 24 | 1 | 1.05 | 1.55 | **0.71** | 0.65 | 0.65 | 0.66 | 0.67 | **0.65** |
| 20 | 88 | 36 | 87 | 6 | 24 | 1 | 1.01 | 1.65 | **0.71** | 0.65 | 0.64 | 0.64 | 0.65 | **0.64** |
| 21 | 96 | 52 | 98 | 67 | 4 | 1 | 1 | 1.14 | **0.8** | 0.67 | 0.66 | 0.71 | 0.7 | **0.65** |
| 22 | 116 | 48 | 134 | 67 | 4 | 1 | 1 | 1.04 | **0.81** | 0.69 | 0.69 | 0.69 | 0.69 | **0.69** |
| 23 | 82 | 31 | 81 | 5 | 6 | 1 | 1 | 1.18 | **0.74** | 0.65 | 0.65 | 0.65 | 0.67 | **0.64** |
| gm | 90 | 34 | 93 | 14 | 8 | 1 | 1 | 4.71 | **0.7** | 0.64 | 0.64 | 0.79 | 0.8 | **0.62** |

Table 4: Summary of results on the 23 datasets (rows). Columns 1 to 5 contain the number of initialization+lloyd runs completed in time limit TL. Columns 6 to 14 contain MSEs relative to the mean initialization MSE of km++. Columns 6 to 9 are mean MSEs after initialization but before lloyd, and columns 10 to 14 are minimum MSEs after lloyd. The final row (gm) contains geometric means of all columns. clarans consistently obtains the lowest across all MSE measurements, and has a 30% lower initialization MSE than km++ and afk-mc$^2$, and a 3% lower final minimum MSE.

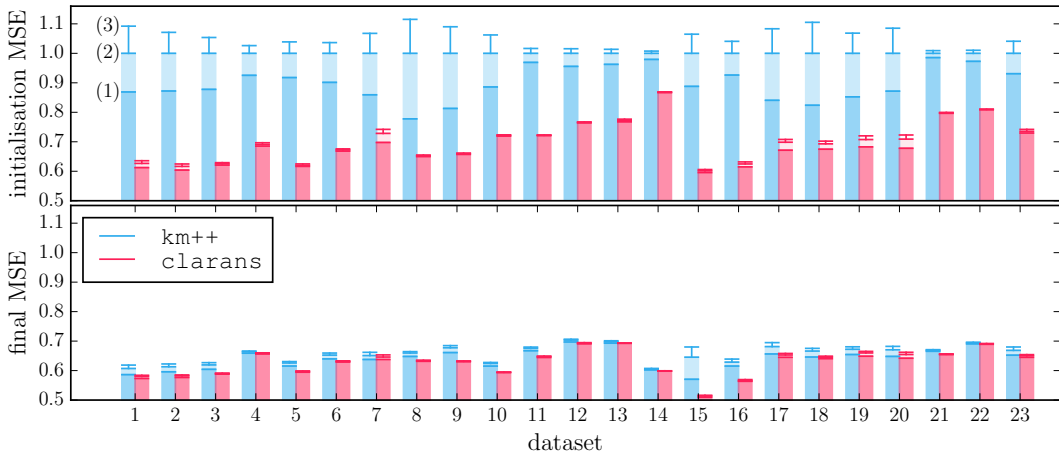

Figure 6: Initialization (above) and final (below) MSEs for km++ (left bars) and clarans (right bars), with minumum (1), mean (2) and mean + standard deviation (3) of MSE across all runs. For all initialization MSEs and most final MSEs, the lowest km++ MSE is several standard deviations higher than the mean clarans MSE.

### 5.1 Baseline performance

We briefly discuss findings related to algorithms `uni`, `bf`, `afk-mc`$^2$ and `km++`. Results in Table 4 corroborate the previously established finding that `uni` is vastly outperformed by `km++`, both in initialization and final MSEs. Table 4 results also agree with the finding of Bachem et al. (2016) that initialization MSEs with `afk-mc`$^2$ are indistinguishable from those of `km++`, and moreover that final MSEs are indistinguishable. We observe in our experiments that runs with `km++` are faster than those with `afk-mc`$^2$ (columns 1 and 2 of Table 4). We attribute this to the fast blas-based `km++` implementation of Newling and Fleuret (2016).

Our final baseline finding is that MSEs obtained with `bf` are in general no better than those with `uni`. This is not in strict agreement with the findings of Celebi et al. (2013). We attribute this discrepancy to the fact that experiments in Celebi et al. (2013) are in the low $K$ regime ($K < 50$, $N/K > 100$). Note that Table 4 does not contain initialization MSEs for `bf`, as `bf` does not initialize with data points but with means of sub-samples, and it would thus not make sense to compare `bf` initialization with the 4 seeding methods.

### 5.2 `clarans` performance

Having established that the best baselines are `km++` and `afk-mc`$^2$, and that they provide clusterings of indistinguishable quality, we now focus on the central comparison of this paper, that between `km++` with `clarans`. In Figure 6 we present bar plots summarising all runs on all 23 datasets. We observe a very low variance in the initialization MSEs of `clarans`. We speculatively hypothesize that `clarans` often finds a globally minimal initialization. Figure 6 shows that `clarans` provides significantly lower initialization MSEs than `km++`.

The final MSEs are also significantly better when initialization is done with `clarans`, although the gap in MSE between `clarans` and `km++` is reduced when `lloyd` has run. Note, as seen in Table 4, that all 5 initializations for dataset 7 result in equally good clusterings.

As a supplementary experiment, we considered initialising with `km++` and `clarans` in series, thus using the three stage clustering `km+++clarans+lloyd`. We find that this can be slightly faster than just `clarans+lloyd` with identical MSEs. Results of this experiment are presented in §SM-I. We perform a final experiment measure the dependence of improvement on $K$ in §SM-I, where we see the improvement is most significant for large $K$.

## 6 Conclusion and Future Works

In this paper, we have demonstrated the effectiveness of the algorithm `clarans` at solving the $k$-medoids problem. We have described techniques for accelerating `clarans`, and most importantly shown that `clarans` works very effectively as an initializer for `lloyd`, outperforming other initialization schemes, such as `km++`, on 23 datasets.

An interesting direction for future work might be to develop further optimisations for `clarans`. One idea could be to use importance sampling to rapidly obtain good estimates of post-swap energies. Another might be to propose two swaps simultaneously, as considered in Kanungo et al. (2002), which could potentially lead to even better solutions, although we have hypothesized that `clarans` is already finding globally optimal initializations.

All source code is made available under a public license. It consists of generic C++ code which can be extended to various data types and metrics, compiling to a shared library with extensions in Cython for a Python interface. It can currently be found in the git repository `https://github.com/idiap/zentas`.

## Acknowledgments

James Newling was funded by the Hasler Foundation under the grant 13018 MASH2.

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
