[Reviews · NeurIPS 2017]

Reviewer 1



The authors propose to use a particular version of the K-medoids algorithm (clarans - that uses iterative swaps to identify the medoids) for initializing k-means and claim that this improves the final clustering quality. The authors have also tested their claims with multiple datasets, and demonstrated their performance improvements. They have also published code that will be made open after the review process. Comments: 1. The paper is easy to read and follow, and the authors have done a good job placing their work in context. I appreciate the fact that the optimizations are presented in a very accessible manner in Section 4. As the authors claim, open source code is an important contribution. 2. The authors should consider the following work for inclusion in their discussions and/or comparisons: Bahmani, Bahman, et al. "Scalable k-means++." Proceedings of the VLDB Endowment 5.7 (2012): 622-633. 3. While the authors have compared actual run times of different algorithms in Section 5, can they also provide some idea of the theoretical complexity (whenever possible)? 4. Related to previous comment: how much impact would the actual implementation have on the runtimes, particularly given that the “bf” algorithm uses BLAS routines and hence can be implicitly parallelizing matrix operations. Is there any other place in the implementations where parallel processing is done? 5. Please number the data sets in Table 3 to make it easy to compare with table 4. 6. In column 1(km++) of Table 4, why is it that km++ does not run unto 90 iterations always given that TL was based on this time? 7. In 5.2, the claim that “clarans often finds a globally minimal initialization” seems a bit too speculative. If the authors want to claim this they must run many many rounds at least in a few datasets, and show that the performance cannot get better by much.

Reviewer 2



This manuscript provides good motivation for why existing initialization schemes for k-means (such as k-means++) are susceptible to getting stuck in local minima and propose the use of a k-mediods initialization scheme named ‘clarans’ (originally published in 1994) for k-means initialization. The authors provide intuition and explanation for why clarans would be a good initializer. The manuscript also details a series of computational optimizations to the clarans algorithm to address the complexity of the original clarans algorithm. The thorough evaluation supports the claims of the authors and the evaluation decisions and details are very clearly explained in the manuscript, making it very easy to reproduce if needed. However, in my opinion, this manuscript lacks any novelty unless (a few of) the optimizations to the clarans presented in the manuscript are new. That does not mean that I think this manuscript does not make a significant contribution. Another concern I have is the following: While the empirical evaluation in the manuscript appears to be quite thorough, a crucial angle missing in the relative performance of clarans is dependence on the number of clusters k. For example, questions such as the following should be addressed (especially in a empirical only paper): does clarans outperform all other initialization schemes for all value ranges of k (on the same dataset) or is clarans better for large k while being at par with k-means++ for small k?

Reviewer 3



This paper studies the utilization of CLARANS method (proposed by Ng and Han in 1994) for k-means initialization. CLARANS is a k-medoids method, which finds clusters with medoids as cluster centers. These medoids can be considered as the most representative data samples, as they minimize the sum of dissimilarities between themselves and their cluster members. The initialization of k-means has been aware of a key to guaranteeing the optimal clustering results. Various approaches have been proposed to find good initialization for k-means. Since k-medoids can find the k most representative data samples, they can be naturally a set of good initial centers for k-means to start. In this paper, authors experimentally show that CLARANS is a better initializer for k-means than other algorithms on 23 data sets, in terms of the minimization of MSE. The paper is easy to follow. The extensive evaluation results support the claim that CLARANS is a good initializer for k-means. However, there are several concerns: 1. The novelty of this paper is marginal. CLARANS and k-means are already well-known clustering algorithms. 2. The study of CLARANS + k-means and comparison with other approaches are mostly from empirical evaluation perspective. The theoretical soundness is weak. 3. CLARANS is computationally expensive. Although different levels of complexities are discussed, CLARANS is difficult to be applied on large-scale data sets for selecting good initial centers. Considering the above weak points, reviewer suggests that the paper should be highly improved before publishing at NIPS.